# Barriers, enablers and outcomes reported by parents engaged with the special educational needs system in England: A qualitative study

Jennifer Saxton[1¤*], Anne-Marie Burn[1], Xinhe Zhang[1], Hilary Toulmin[1,2], Jennifer Parker[1], Helen Casey[1], Jacob Matthews[1], Isaac Winterburn[1], Charlotte Tripp[1], Sarah Barnes[1], Poppy Hall[1], Kristine Black-Hawkins[3], Hayley Gains[1], Tamsin Ford[1] (The HOPE Study)[¶]

1 Department of Psychiatry, University of Cambridge, United Kingdom, 2 North-East London Foundation NHS Trust (NELFT), 3 Faculty of Education, University of Cambridge, United Kingdom

¶ Membership of The HOPE Study is provided in the acknowledgements.
¤ Current Address: University of Cambridge Department of Psychiatry, Herchel Smith Building, Robinson Way, Cambridge, CB2 0SZ.
* jcs230@cam.ac.uk

## Abstract

The UK government is currently seeking solutions to solve the 'SEND Crisis' in England to improve service provision and children's outcomes. Parents play a central role in the identification of their children's needs and support requirements and can provide valuable insight into SEND system functioning. This qualitative study explored the experiences of 22 parents in identifying their children's needs, securing provision, and its perceived impact on the child and family. Participants' children had a range of SEND types, including autism, learning disabilities, and mental health problems. We used one-to-one interviews in conjunction with drawn life 'timelines' to gain a comprehensive picture of participants' experiences of engaging with the system over time. Thematic framework analysis identified legal protections and the advocacy efforts of parents and professionals as key enablers. Barriers included professionals' lack of understanding about SEND (particularly autism), poor communication between services and families, and system failures. Positive outcomes parents attributed to SEND provision included health, education and social improvements, and young people gaining autonomy and independence. Perceived negative outcomes included lost educational opportunities, worsening mental health for children and their parents, and educational policies lacking long-term vision. We provide narrative portraitures which echo these themes. Our study suggests that solving the SEND crisis will require multiple actions, not only to repair fractured relationships and improve communication between professionals and families, but standardisation of key processes to reduce unfair variation in provision. The current system heavily depends on advocates within it, which can inadvertently harm parents who engage with it; children without advocates are at risk of missing out on much needed

**Data availability statement:** The datasets generated and/or analysed during the current study are not publicly available due to the risk of directly or indirectly identifying participants from their special category data (e.g. participants' health status or social care involvement, and their children's special educational needs and disability status). This small and specific participant group, and their detailed interviews which included life stories over time, mean there would still be a risk of inadvertently disclosing participant identities if we released whole transcripts for external access - even if transcripts were anonymised by standard procedures. To protect participants from these risks and to maintain confidentiality, the dataset remains closed. For external requests or enquiries regarding data access, please contact the University of Cambridge's Data Management Team at: https://camide.cam.ac.uk/ (The University of Cambridge Integrated Data Environment). The data are held on a secure drive inside password protected folders within the CAM:IDE infrastructure. Requests will be reviewed to ensure they comply with ethical and data governance requirements.

**Funding:** This research was supported by Health Data Research UK under grant No. LOND1), which is funded by the UK Medical Research Council and eight other funders. TF is supported by a National Institute of Health Research (NIHR) senior investigator award (https://www.nihr.ac.uk/). All research at the Department of Psychiatry in the University of Cambridge benefits from the NIHR Cambridge Biomedical Research Centre (NIHR203312) and NIHR Applied Research Collaboration East of England. This study/project is funded by the National Institute for Health Research (NIHR) under its Programme Grants for Applied Research Programme (Grant Reference number NIHR- NIHR202025). The funders did not play any role in the study design, data collection and analysis, decision to publish, or preparation of the manuscript. The views expressed are those of the author(s) and not necessarily those of the NIHR or the Department of Health and Social Care. There was no additional external funding received for this study.

**Competing interests:** The authors have declared that no competing interests exist.

provision. Our study also demonstrates that provision can substantially improve children's health, education and belonging in society. The government's goal should be to ensure that this level of effective provision is accessible for all children with SEND.

## Introduction

### Background

**Special educational needs and disabilities (SEND) policy context.** Special Educational Needs and Disabilities (SEND) services in England are facing increasing scrutiny. The UK's public spending watchdog has determined that the system is failing to meet the support needs of many of the 1.6 million children with SEND, offers poor value for money and is financially unsustainable [1–3]. The system was overhauled in 2014 [4] to ensure earlier identification of children with SEND, longer support throughout education (up to age 25), and to strengthen families' involvement in decision-making that affects them. Other changes included new legally enforceable education, health and care plans (EHCPs) for children with more complex needs overseen by local authorities (LAs) with integrated inputs from health, education and social care sectors, and additional routes of redress if families wished to appeal against LA decisions. This year, the government asked for solutions to solve the 'SEND crisis' [5] and is engaged with parallel reforms to curriculum and assessment, children's wellbeing and disabled children's social care law which will directly affect children with SEND [6–8].

**Evaluations of the SEND system.** More than a decade since the 2014 reforms, evidence from appeals and complaints data suggests that dissatisfaction of families has increased, with 98% of parents successful in their appeals to First-Tier SEND tribunals [9,10]. Educational data on the impact of SEND provision on children's outcomes is of particular concern, including lack of improvement in educational attainment and disproportionately higher school exclusions relative to peers without SEND [11,12]; young people with SEND are also more likely to be 'not in education, employment of training' (NEET) [13]. Geographical and socio-economic variation in educational outcomes suggests variable implementation and inequitable coverage of services [14,15]. SEND legislation has been criticised for being too narrow and under-resourced, lacking clarity about processes, being vulnerable to wider negative political and economic influences, and overly outcome focused for SEND learners in mainstream education settings [16–18]. Parents/carers of children with SEND (hereafter termed 'parents') are key stakeholders, central to children's SEND identification and securing their children's provision. As advocates for their children and directly engaging with the SEND system and the professionals involved, they can offer valuable insights into how the SEND system can be improved to have a positive impact on SEND learners.

**Systemic and socially determined barriers experienced by parents.** Qualitative literature from parents in the UK indicates systemic barriers to families who attempt to access support for their children; children are further disadvantaged when parents are no longer able to engage with the system [19,20].

In one study with parents from a UK inner-city borough, with a diverse and high-density population, respondents reported exhaustion from having to battle against systemic discrimination necessary to secure SEND provision for their children. Their experience was that support was provided reactively when children are withdrawn, rather than being provided proactively from an integrated education and health care system [21]. The study authors suggest their evidence points to a systemic issue, requiring multi-systemic solutions such as that depicted by Bronfenbrenner [22]. Another study found evidence of disproportionate removal of services for children with SEND compared to those without during the COVID pandemic [23] which further illustrate socially determined patterns of service provision that are associated with inequitable outcomes.

**Education-sector specific barriers experienced by parents.** Several studies have focused on the link between families and the education sector, with adverse relationships with school staff often reported, and many respondents questioning the adequacy of professionals' training in identifying and providing for SEND. One study reported that SEND guidance was poorly understood by schools, and that parental experiences of inclusion in mainstream settings were overwhelmingly negative. Families experienced discrimination, a lack of diversity and inclusion awareness from others, a need to advocate for their child and seek legal advice, a decline in well-being and mental health, economic decline and damage to family relationships [24].

Our own research, involving an online survey of key stakeholders in SEND identification and provision identified that more than two-thirds of parents disagreed that education staff were sufficiently trained to work with children with SEND, and around half reported their overall experiences of SEND identification and provision as negative [25]. In related survey work [26] we found only a quarter of parents believed education providers had good or very good understandings of their children's support needs, with fewer still endorsing the understanding of health and LA professionals; only half believed that their child's future goals and aspirations were adequately represented in their EHCP. Other studies have found that teachers can reduce their aspirations for children who are identified as having SEND [27]. One commentary published at the time of the SEND reforms encouraged the education sector to move beyond the grief model emphasising deficits of SEND towards strength-based approaches to empower and meaningfully engage and collaborate with parents [28].

**Support through key transitions.** Some research indicates that children's vulnerability at key transition points is not always sufficiently recognised and provided for, and communication with families can become patchy. The transition from primary school to secondary school is a specific area of potential vulnerability. Parents and schools have an important role in helping children settle into a new environment, but young people with SEND sometimes continue to experience difficulties after the transition. Interviews and focus groups with parents in Wales found that before transition, parents experience a loss of primary school community support and a fear that hard-won gains may be undermined. They experienced a lack of communication from schools about the transition process and a hesitancy to petition schools for help in case children are perceived as a burden to the school. After the transition, parents reiterated the importance of social support in addition to the efficacy of early support and good transition planning. However, they also experienced a continuation of poor communication and lack of information [29].

**Parents' preferences and involvement in decision-making.** From the quantitative literature focused on educational placements, a small survey of parents identified that the top three factors influencing their choice of special school placement were school atmosphere, a caring approach to pupils and class size [30]. Despite new obligations that arose from the SEND Code of Practice 2015, including parent-professional partnerships in designing provision, research has identified conflict between professionals and parents about placement decisions and other aspects of service provision that are difficult to resolve. Whilst the SEND Code of Practice encourages parents' preferences it affords them limited decision-making power, and key decisions are taken by professionals, often constrained by resources and what the professional deems to be the most suitable provision [31].

**Reported outcomes following SEND provision.** Much of the small literature that has consulted parents and examined outcomes following SEND provision identifies that engaging with services can have adverse impacts on the

mental, physical and financial wellbeing of the whole family [19,20]. For parents, these outcomes are often driven by the requirement to persistently fight for and privately fund assessments to access SEND-related treatment, and parents working in isolation in an intimidating system to obtain support for their children. There are also inequitable outcomes between different groups of children with SEND. For example, a scoping review of evidence from the UK looking at evidence of any impact for children with SEND during the COVID lockdown period, found greater risk of mental health problems in single parent families, those from deprived neighbourhoods and ethnically diverse parents [32].

### Rationale

Whilst several studies have reported on the negative experiences and consequences for parents navigating the SEND system, few peer-reviewed studies have examined the link between SEND provision and children's and families' health, education and other outcomes, or the antecedent barriers and enablers of identifying SEND and securing appropriate provision. Neither could we find any peer-reviewed studies that explore whole-system experiences from a child's SEND identification to provision and into adulthood from parent perspectives. Given that greater involvement of parents in SEND-related processes and decision-making were central to the 2014 reforms, and they and their children are in some respects casualties of the current system, it is pertinent that we hear from parents now to shape reforms that are underway.

Our qualitative study aims to improve our understanding of parent-reported enablers and barriers to SEND identification and provision, and how they influence different outcomes in the England context. These findings will feed into the current solution-focused SEND enquiry.

We seek to answer the following research questions:

1. What enablers and barriers do parents experience in identifying Special Educational Needs (SEN), and securing and receiving SEND provision?

2. What health and other outcomes do parents report as a consequence of their child's SEND provision?

## Materials and methods

### Ethical approval

We secured ethical approval for the study from the Cambridge University Psychology Research Ethics Committee (PRE.2021.058).

### Study design

This qualitative study involved one-to-one interviews structured around a participant's 'timeline' from the birth of their child, through their child's education to date, and looking ahead to their child's future. We drew out timelines acting as scribe, under the guidance of participants, adding in brief written details and drawings based on key events and turning points they described. The use of timelines in qualitative research helps to elicit complete participant accounts depicting key events from participants about their experiences over time [33,34]. Given the complexity of the SEND system and that parents were likely to have a rich and detailed account to share with us spanning years, we considered in-depth interviews to be the most appropriate data collection method to answer our research questions. Our manuscript adheres to the consolidated criteria for reporting qualitative research (COREQ) [35] (see supplementary file S1 File).

### Interview topic guide development

We developed a semi-structured interview guide (see Supplementary file S1 File) based on our previous SEND survey of parents and input from our parent advisory group to finalise content and question wording. We pilot tested the guide within mock timeline-based interviews with four parents of children with SEND (two online and two in-person), and the data

collection team conducted several practice interviews with each other until they were familiar with the schedule and the supporting technology to draw timelines and record the interviews.

**Data collection team and reflexivity.** Five researchers collected the data (3 female, 2 male). The team all had working and/or research experience with children with SEND and their families. The team included one post-doc, two research assistants, and two second year psychology undergraduate students. The team were supervised by three senior researchers, all of whom had considerable research experience using relevant methods to explore sensitive topics. Two of the senior researchers designed and carried out a half-day workshop to train the data collectors in conducting interviews, including how to ask open questions, the use of timelines, and safeguarding. The workshop also involved role-plays, and senior researcher feedback to data collectors based on their pilot transcripts. The data collection team continued to practice interviewing with each other until they were confident in using the methods and following the data collection protocol.

## Participant recruitment

We recruited participants from a list of our previous survey respondents who said they were happy to be contacted about future research. We purposively aimed to recruit a mixture of participants by region of England, EHCPs and SEN support, and children's need types. Participants were approached by one of the data collection team by email initially, referencing their previous participation in the HOPE Study survey, and that we were motivated to find out about their experiences in more detail. People who said they wanted to take part were followed up with telephone calls or emails to arrange interviews. We emailed participant information sheets and consent forms to those who expressed an interest. Participants provided acknowledgement of the participant information sheets and written informed consent, via signed consent forms, which they returned to us by email prior to interview.

We considered that 20–25 participants would be sufficient to achieve information power [36]. We oversampled the number of parents we invited to interview on the assumption that not all would agree or end up being able to take part.

## Data collection

Data collection took place between May and July 2023, at participants' homes and online. Interviews lasted between one to 2.5 hours (median = one hour 20 minutes). There were no repeat interviews. For researcher safety the team went in pairs to in-person interviews, whereas online interviews involved one researcher only. Interviews were audio recorded either on the study iPad or using Zoom software. Timelines were drawn using pen and paper for in-person interviews and using Scribble Together Whiteboard software for online interviews and were used as an aide-mémoire and to generate discussion. The data collection team made field notes after each interview for reflective purposes.

## Data analysis

Interviews were transcribed verbatim and then anonymised before analysis. Transcripts were not returned to participants and we did not ask them for feedback about the findings. Five of the study authors (JS, AMB, XZ, HT, JP) coded the transcripts using the thematic framework approach [37,38]. Gale et al's method follows the systematic stages of 1) transcription 2) familiarisation with transcripts 3) coding 4) developing a working analytical framework 5) applying the framework 6) charting the framework into a matrix (participant rows, thematic columns) 7) Interpreting the data. The method seeks to combine inductive and deductive approaches to generate themes from the data – the former based on people's own accounts, the latter using existing literature and the topic guide questions. Gale et al also provide an applied real-world example demonstrating its suitability for use by multi-disciplinary teams and for policy research (ibid). First, we deductively coded a subset of transcripts using our two overarching research questions, while also allowing for inductive coding to capture meaning generated from the data. Team members met to discuss the coding and to develop the analytical framework which was adapted through a series of coding rounds. The framework and transcripts were transferred into NVivo

V14 for coding. The resulting coding matrix containing relevant coded text for each theme by individual case was exported into Excel. Coders were then assigned themes and subthemes from which they produced thematic summaries (charting the data) for each individual participant, and then as an overall summary memo for their assigned theme and subthemes, while also retaining pertinent verbatim quotes to illustrate key themes. The final coding framework used to generate participant and thematic summaries is available in the Supplementary file. Two study authors then derived the final streamlined set of themes and sub-themes reported in this paper at the last stage of analysis – this was to highlight strong areas of congruence and contrast between themes and eliminate repetition when all the summaries were available.

In parallel, two study authors developed narrative portraitures from three full transcripts. We selected three transcripts that we thought strongly echoed the themes we had identified. Narrative portraits are complete presentations of participants' narrative accounts, that respect individual stories, provide complementary balance to thematic analyses which combine individual stories, and are readily developed from timeline-based interviews. We followed recent guidelines [39] to develop the narrative portraits, annotating the transcripts using different colours to identify: 1) key 'characters' 2) time scales; 3) sentences that provided spatial orientation in geographical, domestic and other spaces (such schools) and virtual spaces (online, emotional) 4) key events and turning points and how they inter-related with other factors to answer our research questions about barriers, enablers and how they linked to SEND provision and outcomes. Using these marked out portions of narrative, in conjunction with characters in time and space, we constructed the narrative portraits.

## Results

### Participants

In total we carried out 22 interviews with parents/carers; 20 were conducted online and two were held in person at the participants' homes. 70% of the parents were female and were mainly biological parents; a minority of respondents were foster and adoptive parents. The median age of participants' children was 14 years (range 4–24 years). 9 children were male and 13 were female. The most recent education settings for children were home-education (n = 3), independent school (n = 3), mainstream settings (n = 6), specialist schools or colleges (n = 9) and university (n = 1). Nineteen children had EHCPs or a SEN statement (both are legal documents outlining plans for children requiring more support than schools ordinarily provide). Three children were receiving 'SEN support' (lower-level support, largely led and provided for by schools). The parents of children receiving SEN support were privately funding SEND-related help outside of school or were awaiting an EHC assessment.

Many children had multiple additional needs (median = 3, min = 1, max = 8). Parents described their children's primary and secondary needs to us as follows: Autism (n = 7/22), Attention Deficit Hyperactivity Disorder (ADHD; n = 15/22) cognitive support needs (n = 9/22), speech and language difficulties (n = 9/22), hearing impairment (n = 4/22), visual impairment (n = 1/22), multi-sensory impairment (n = 2/22) physical disability (n = 6/22), chronic health conditions (n = 3/22), and other SEND (n = 8/22) which included mental health problems, brain injuries, and specific syndromes.

We identified four themes which are summarised in Table 1 and illustrated in this section with participants' quotes.

### Theme 1: Enablers of SEND identification and provision

**Parents as advocates for their child.** Parents knew their children's needs well and were usually their main advocate in securing SEND provision, including changing education settings. They described having to speak up against decisions they disagreed with and often used military language such as "fight" reflecting the hostility of their experiences within the SEND system.

*"Right from the start of our journey, it's been a battle."* (#201)

*"…then I had to fight."* (#220)

**Table 1. Overview of themes and subthemes.**

| Main themes | Sub themes |
|---|---|
| Enablers of SEND identification and provision | • Parents as advocates for their child<br>• Professional advocates/ champions<br>• Legal protections |
| Barriers to SEND identification and provision | • Children can be misunderstood and unheard<br>• Poor communication and relationships between professionals and families<br>• Systemic failures and design flaws |
| Positive outcomes following SEND provision | • Improved educational, social and health outcomes<br>• Gaining autonomy and independence |
| Negative outcomes following SEND provision | • Child mental health crises from delayed, inconsistent or prematurely removed provision<br>• Lost education from slow or inappropriate decision-making<br>• Education policies undermine children's health and academic potential<br>• Professionals' lack of aspiration and long-term strategies, limiting children's potential and independence<br>• Parents' mental health, capacity to work and family finances are harmed by the system |

There was evidence that parents needed to have exceptional organisational and administration capacity to successfully obtain SEND support for their child within a complex system. They also had to be well-informed about their child's legal rights to provision, be able to understand complex terminology and take on the role of proactive experts to advocate effectively:

*"It was all sorted out by us. Nothing from the school." (#241)*

Many parents were also covering the financial cost for support, such as therapies to support children's mental health and additional professionals' reports required for legal processes. They voiced concern that children without parents or carers to advocate on their behalf, or who lacked financial resources, were unlikely to receive the support they needed.

*"The parents are the ones who can, can make or break with a kid's education, depending on how you know, wealthy they are, and what income they've got coming in." (#209)*

*"She either has no counselling, or, no help for her mental health, or I pay for it." (#236)*

Some parents chose to home-school their children, as it allowed greater flexibility to obtain support that was better tailored to their child's needs.

*"I could meet his needs, and that's the first time every aspect of his EHC has been met when I've done it myself." (#201)*

**Professional advocates/ champions.** Several parents spoke about unofficial advocates within the health system who helped facilitate SEND assessment and provision, easing what was otherwise a difficult process for families. Paediatricians were frequently described in positive terms, for example, one parent shared that their child's paediatrician aided an ADHD assessment, despite the child being referred for autism. Another mentioned that a child and adolescent mental health services (CAMHS) psychiatrist advocated for her child at annual reviews (#201), and another spoke about being listened to and supported by speech and language (SALT) and occupational therapists (OT) (#234). Within social care, one parent said:

*"The first social worker she had, she was a bit of a [deep breath] ugh I don't know, she just wasn't very proactive, but then the second one she had, he's, he's wonderful …he's lovely, he, he's always fought [CHILD NAME]'s corner and always been very kind and patient and, yeah just wonderful. So we, you know, we have been really blessed with the people that have been in [CHILD NAME]'s circle as it were." (#227)*

Charities such as the SEND Information, Advice and Support Service (SENDIASS) were instrumental in guiding and advocating for parents as they navigated the SEND system, and playing a key role in holding authorities to account:

*"The SENDIASS team in [LA] is very, very helpful. And they are very, very, very thorough with the knowledge they have and the guidance they give." (#207)*

*"…she was like where's his termly support plan, and they said ohh we haven't done one yet. Well it's six months into the year like why haven't you done one, and she's, like and then, you know, she just kind of like came in like a cannon ball sort of thing and just like blew everybody away." (#226)*

**Legal protections.** Protected status and characteristics were identified as important enablers of access to SEND provision. One parent thought the lack of barriers they had experienced in securing SEND provision was largely due to their child's looked after status, which provided additional statutory protections. As a result, they received excellent social care support, including regular personal education plans (PEPS), care reviews and other strong service provisions:

*"A flag in the system means she doesn't have to wait or struggle to get health appointments – including CYPS [formerly CAMHS] …we have social workers, and we have an advocate and we have a supervising social worker because we're fostering…they have a statute of responsibility to, to make sure that she's well, looked after, safe. And also that we are as well." (#211)*

Other parents drew on The Equality Act (2010), specifically the legislation on reasonable adjustments for disabled people, to hold schools accountable for delivering appropriate SEND-related provision:

*"They are not up to the teacher to decide whether they're done or not. And that it's a reasonable adjustment to have a sensory aid…which were denied…They were told not to fidget, they were told they couldn't have movement breaks." (#210)*

*"People are very good at following the law of reasonable adjustments as far as physical goes." (#247)*

Redressal routes, though often slow and costly, were described as highly effective in securing SEND provision. Around half of the participants had legally challenged LA decisions through tribunals (sometimes more than once), judicial reviews, or by contacting their MPs when LAs became unresponsive. The most common reason for pursuing tribunals was to ensure children could attend an educational setting capable of meeting their SEND needs, as LAs named settings that were frequently unsuitable. Despite universally successful legal outcomes, the legal processes were time consuming and led to some children being out of education for extended periods. These delays caused parents significant distress and financial expense. One parent noted that their tribunal was "fairly easy on the day" because the LA had come unprepared:

*"They didn't have any, any settings where he could live, they didn't have any work for him, they had nothing they could offer in place of a college setting….we had a legal advice and you know, solicitor on our side, who had done all the sort of work for us. We'd got an educational psychologist's report and a speech and language report and all that kind of stuff that nobody's bothered to do, for years." (#224)*

*"[The LA] would only consider putting her into another mainstream school, even though the head of the first mainstream school that had excluded her had said she needed to be in a specialist environment…we employed an SEN solicitor, threatened tribunal and they caved in a week before tribunal." (#236)*

One parent shared that the LA "buckled" when they threatened legal action over a placement (#209), whilst another explained that when their application for an EHC assessment was refused:

*"…within a week of submitting the appeal papers, they, they emailed to say that they'd reviewed their decision and were going to assess." (#247)*

### Theme 2: Barriers to SEND identification and provision

**Children can be misunderstood and unheard.** Professionals sometimes lacked understanding about SEND and/or had limited knowledge of the child concerned. Several parents reported significant challenges with their children's schools, often stemming from school staff attitudes and lack of understanding or knowledge of their child's condition (particularly autism) and specific needs. Parents recounted instances where school staff reprimanded children for behaviour that was directly related to their SEND, such as fidgeting, rocking, or avoiding eye contact. These responses caused their children emotional distress and, in some cases, lasting "trauma". One parent described how their autistic child was placed in a group activity aimed at improving social skills. However, this only seemed to reinforce "ableism", as the focus was on forcing the child to "fit in" rather than addressing their individual needs. In some cases, misunderstandings about children's needs led to children being excluded from class and prolonged absences from school.

The level and quality of support from teachers varied widely, often depending on individual staff members' understanding of SEND and their willingness to engage with the child's needs. Teachers with formal SEND training or personal experience of autism tended to show greater empathy and provide support, whereas others failed to follow through with strategies or provide consistent care. Similarly, parents noted that the support from teaching assistants varied depending on their level of training and understanding. Several parents expressed frustration with schools' inability to consistently recognise and accommodate their child's needs, observing that when needs were overlooked, children experienced increased emotional distress and escalating behavioural issues. In one case, a child who had successfully used an iPad for communication was no longer allowed to use it after transitioning to college, as staff reportedly found it too difficult to manage.

There were many instances where the level of support provided by schools did not meet the parent's expectations and they reported that the lack of proper training for special educational needs coordinators (SENCOs) was a barrier to providing effective support. Frequent changes in key staff, such as the SENCO, were reported to disrupt continuity and hinder their child's progress. Some parents also raised concerns about the qualifications and expertise of staff in specialist roles, including one case where a behavioural specialist was reportedly not qualified for the role.

Misunderstandings of autism and other neurological differences (such as ADHD), and the different ways they can present, also affected the identification of SEND and access to support within the health system. This was evident in referrals and the advice parents received during appointments. For example, one parent whose child presented early social difficulties was told by two general practitioners (GPs) "she will outgrow it" (#234). Another parent reported being told by an educational psychologist (EP) that her child was "not autistic enough" to meet the criteria for services (#236), and a neurologist told another parent that nothing more could be done to support their child (#205). Several parents expressed dissatisfaction with how CAMHS handled their cases. In one instance, after assessing an autistic child at school, CAMHS staff reportedly blamed the parent for the child's behaviour and said to the child:

*"…you're a very naughty child, you're attention seeking, stop self-harming and trying to kill yourself." (#236)*

Another child who had been accepted and treated by CAMHS was discharged while still in crisis (#210). In another case, a parent reported being dissuaded by a SENCO from reapplying for CAMHS treatment for her son. She was told he was "too medical" as he has a brain injury and the referral was likely to be rejected (#220).

Misunderstandings of autism within the social care system similarly affected SEND provision and resulted in negative experiences for families. Several parents noted that Pathological Demand Avoidance (PDA) was poorly recognised or understood by social workers and professionals within the criminal justice system.

> "…even on that disability team, they didn't bloody understand autism. They didn't understand ADHD, and they, you know they just don't get it." (#201)

> "…if there's an element of SEND, don't send a social worker who has no clue about SEND, because that's damaging, it was damaging for us." (#226)

SEND-related characteristics and children's communication difficulties meant that children's needs were sometimes unheard. Parents described how children with delayed or disordered speech and language faced inherent communication challenges, which made it difficult for them to communicate their SEND-related needs. This undermined the assessment of their needs, leading to an inadequate level of provision:

> "We started thinking something's not right here. Again, just got swallowed up because she was really quiet…She just got overlooked, quite a lot." (#228)

Likewise, children with autism often struggled to articulate their needs. Parents noted that their children would engage in "masking" to suppress or hide their feelings, which further complicated the identification of their needs, which in some cases, became more pronounced over time:

> "…they were much better at voicing in their needs in their early years than throughout school because shutdowns started to happen around year three." (#210)

Several parents highlighted that their autistic children were unable to initiate interactions to express their needs. This often went unnoticed by the adults responsible for their care, and which negatively affected their school experience, attendance and overall outcomes.

> "The Ear Defenders was the huge one, they'd keep them in a drawer in the teacher's desk and say he can ask them whenever he wants and I'd say but he won't ask for them, like we can go out and there can be a huge noise, and he can just collapse on the floor and start screaming, and I know to just give him his ear defenders, I don't say would you like your ear defenders, because he's not able to communicate and I felt like they just weren't able to get that."(#226)

Autistic communication styles often involved children responding with direct honesty, which was sometimes misinterpreted as rudeness. These misunderstandings could strain relationships with education staff, negatively impacting both the support provided and the child's emotional wellbeing:

> "…He was always pulling people up. But the teachers used to just see him as being cheeky, cocky. And they would take him on head on. And then they just bring the worst out in him." (#201)

Communication difficulties sometimes meant that parents were unaware of the support that was being provided at school. This lack of transparency contributed to mistrust toward education professionals, and left parents unsure how to contribute towards their child's support.

*"…it's possibly easy to hide things like that with, with students who struggle with communication, who can't really say what's happened in the day.' (#224)*

Some parents reported that SEND provision had been harmful for their non-verbal children. One parent discovered that their non-verbal child with mobility difficulties was not being taken to the toilet at school as previously agreed. Instead, staff had decided – against the child's and parent's wishes – to use nappies, which led to the child becoming extremely distressed and incontinent. Another parent described how the teaching assistant had made their child's situation worse:

*"My child was better regulated when they didn't have the teaching assistant involved because the teaching assistant, later on I found out, made fun of my child. Didn't have any insight whatsoever into autism and he thought that any behaviour, any impulsivity, and eagerness to talk was showing off or came from being, I don't know, rude or something" (#210)*

**Poor communication and relationships between professionals and families.** Many parents reported inconsistent, inadequate, and dismissive communication from schools, health services, social care and LAs. For example, sometimes there was resistance at school-level to implementing recommended support, forcing parents to fight for appropriate support because their child's needs were not adequately met. Families found themselves navigating a complex system alone.

Parents emphasised that effective communication was key to meeting children's needs. Clear, consistent, and collaborative communication was particularly valued for making meaningful changes and adjustments. For example, some parents would like regular email updates from schools to help them understand their child's daily experiences and progress. However, many described their relationships with professionals as characterised by low trust and poor communication, which in turn undermined children's provision.

Within social care, several parents reported that professionals struggled to differentiate between cases of safeguarding and situations involving children with SEND who were not at risk of parental harm. This negatively affected SEND provision for children and damaged relationships between families and social services. One participant had been subject to safeguarding investigations by social services:

*"It was crazy. And stupid and pointless. And I still don't know, actually, there was an investigation into me. And nobody still got in touch with me all these years later and told me what's happened about that…But there must be a report and an investigation somewhere… social services don't seem to want to communicate with anybody." (#221)*

Another parent said they were only taken seriously about needing support after her child stabbed her, and then child protection enquiries were:

*"…dished out and you know and your life is made to be a misery." (#201)*

Relationships between LAs and parents were described as particularly poor. These difficulties were often the result of patchy or non-transparent communication, unclear decision-making, and a tendency to blame parents – all of which contributed to delays in children receiving provision. One parent said the relationship with the LA broke down when their child was due to transition to high school and the EHCP caseworker was unresponsive:

*"…terrified…the caseworker wouldn't reply to emails, wouldn't reply to voice conversations, wouldn't reply to the school, wouldn't reply to ISCA…the school have accepted him on an ECHP that's out of date, the fact that he's got to wear a helmet…there's all sorts of restrictions on his movement." (#220)*

Interagency communication was sometimes poor, which impacted upon the quality of communication to families and in some cases children would have been left without education placements unless parents took extreme steps to intervene.

*"I…contacted our MP and said…'she's supposed to be starting college in two weeks and, we've had nothing…the place you know, it has been agreed, but there was nothing'….her, caseworker I guess is the word, at the Special Educational Needs department…he never answered an email. He never came to a meeting. He didn't even answer the social workers emails, never mind mine." (#227)*

Outdated and poorly written plans reflected a lack of communication and missed opportunities with children and families to ensure appropriately designed support. Legal challenges and other complaints and disagreements with LAs were often driven by poor quality support plans.

*"It wasn't specific, it wasn't quantifiable...cut and pasted from when he was three years old, it wasn't up to date, it was absolute rubbish." (#201)*

Several parents noted a lack of input into plans and reviews from autism specialists which meant plans and provision were less tailored. Some had negative experiences working with SEND workers on EHCPs, which undermined their children's provision:

*"[The SEND worker said] We'll update the EHCP [because it] is a document that can grow with [CHILD'S NAME]. Well it was a lie, because it never grew with [CHILD'S NAME], it never changed. There were a couple of things that changed. But it was never, it was never to the extent that it needed to be changed." (#220)*

**Systemic failures and design flaws.** Many parents reported being excluded from multiagency decision-making processes. Rather than being active participants, they were often on the receiving end of decisions that went against the best interests of their children, and which they often felt powerless to challenge in the short-term. This worsened relationships between families and professionals, leading to delays or prematurely removing vital provisions.

*"How can they close cases and say 'everything's okay, your child is alright', when they are suicidal or self-harming, it's beyond, beyond!" (#210)*

Adding to the sense of powerlessness and feeling misled about their child's provision was the experience of being ignored or not being able to contact anyone for clarity.

*"Because I can't speak to anybody, because nobody will answer." (#227)*

Other systemic failures, partly driven by variable and high thresholds for care, referral problems and waiting lists also affected SEND identification and provision. Several parents said thresholds for starting or maintaining support within social care were unreasonably high to "qualify", despite clear evidence of need. Parents described feeling as though they were being bounced from one provider to another and they commonly reported waiting times of over two years for CAMHS, SALT and physiotherapy. In some areas, services were "non-existent" unless you paid privately (#247).

LAs were also reported to have inconsistent definitions of different types of SEND and varying criteria for what they recognised as eligible for additional support, thus undermining SEND identification and provision, and equity across local areas.

*"…you need to get a dyslexia assessment [but the EHCP reviewer subsequently said] 'dyslexia is not a disability, we do not consider it in this local authority. You may fund your own assessment'. I cannot pay 800 quid, I'm sorry…I don't have the funds…So I'm a ball and they are playing ping pong." (#210)*

Furthermore, inflexible LA-level policies about when children could be assessed added to delays in children's provision. In reference to the compulsory "graduated approach" one parent said:

*"The local authority require us to, to look at, three cycles before they allow us to submit a plan, for a plan. So she said, there's nothing we can do until, well basically, till there had been three half terms. So that was going to take us into, end of March." (#231)*

Several parents noted that services appeared to be at loggerheads with each other about service eligibility, funding, referrals and children's needs, which affected children's provision:

*"Social workers and special educational needs are not working together, at all. They're working against each other…at residential college, part of that is funded by social services, part of it is funded by special educational needs so they're fighting amongst themselves about who has to pay." (#235)*

Many parents thought LAs based their decisions about provision on funding as opposed to children's needs, and that they were not accountable when they did not uphold SEND law:

*"[The LA SEND team's] whole decision-making process is based around funding…there's not unlimited money but I feel like sometimes the decisions that have been made for [CHILD'S NAME] aren't always about what's best for him." (#226)*

*"…when you've got a child with a complex disability…it shouldn't be harder to battle the system than to actually deal with the disability…it should be seamless, they should stick to the law. And there should be accountability all the way through it." (#201)*

Many parents told us about commissioning problems and funding instability negatively affecting their children's provision. Inadequate funding and commissioning of SEND services was said to reduce the quality and continuity of provision. Charity funding was known to be unstable, and clubs that supported SEND sometimes closed if they were not subsidised by LAs:

*"The problem when you rely on charities, if charities run out of money because it's hard times, what are you left with? Nothing. So, since January, my child is clumsier again…that charity may not have closed or may, may be able to reopen if they had the support from, from a local authority. But, they are not interested in commissioning any therapies." (#210)*

Finally, lack of specialist and specialist services/settings (partly due to commissioning failures) were an LA-level barrier to good SEND provision. For example, one parent noted there was no SALT available in the LA which their child needed:

*"The local authority they didn't have any anybody available to be there as much so, I think that meant that there was less attention on the speech and language in the school." (#224)*

Many participants reported that they system did not adequately prepare children for key transitions, particularly the move between primary and secondary school, and from childhood to adulthood. When combined with poor communication

about the child's needs, these adjustments were particularly difficult. In some cases, there was a gap in support across different school years, usually due to unprepared or untrained staff who lacked an understanding of SEND. Parents frequently expressed concerns about the challenges their children faced when transitioning to adult health and social care services, and the lack of continuity in SEND provision after the age of 18. One participant likened losing your paediatrician when transitioning to adult health services as being like "losing your team captain" (#221). Another said their child "got to the age of 18 and fell off the big adult services cliff" (#236). Some were worried about the transition to adult social care and their child's vulnerability as they entered adulthood:

> "…an independent advocate for [CYP NAME] would be great…she's got admirers but she's vulnerable, you know, I worry whether she'd, whether she'd be in a coercive behavioural type situation…if support is not there, she'll follow the wrong crowd… she wouldn't know she's being taken advantage of." (#211)

### Theme 3: Positive outcomes following SEND provision

**Improved educational, social and health outcomes.**  Parents linked tailored support and meaningful engagement with improved educational outcomes for their children. Smaller class sizes had the benefit of reducing sensory overload and fostering a more supportive learning environment. One child made progress in English and maths within the structured environment of a special school, where small class sizes with lower pupil-to-teacher ratios supported learning, while another child thrived in a child-led home education setting, demonstrating increased confidence and engagement:

> "She's doing really well with her reading, really well with her learning actually across, across the board. Attendance, she's improved quite a lot." (#247)

Providing social and emotional support activities in education settings, such as music therapy, aromatherapy and other therapeutic interventions, was seen as beneficial in addressing children's mental health needs. Another perceived advantage of specialised settings was the involvement of professionals, such as occupational therapists, physiotherapists and SALTs, who were equipped to address a child's complex needs. However, schools were sometimes financially constrained in being able to offer comprehensive services. In contrast, some children thrived in child-led home education or Montessori-inspired approaches, demonstrating increased confidence and social engagement when their individual needs were recognised and met.

> "10 weeks of sensory integration therapy as part of his EHCP…was mega useful for him. But he sees his psychotherapist as well. And just the home environment. So all those things together kind of, have improved his health and well-being and his overall kind of happiness…his ability to recognise and communicate his needs and also to kind of self-regulate." (#226)

SEND provision was frequently discussed in relation to children's social outcomes. It often played a key part in developing meaningful social relationships, particularly in settings that offered small groups or structured activities, such as music sessions, which helped foster peer connections and reduce feelings of isolation.

> "It boils down to being equal with her friends and being able to walk with them and, you know, reality if she's in a chair, she's at the back with a teacher, whereas she can walk with a partner along the way…when you're eight, it's really important to just be like everybody else kind of thing. And so for her, it's, it's made her more confident in herself." (#247)

Charities played an important role in enriching SEND provision for home-schooled children, enabled participation in society and enjoyment of life beyond their education, and provided respite for parents: One described charity support as

"a lifeline." (#210). Community-based activities, such as theatre and sports, provided alternative development and emotional regulation. Engagement in activities like acting, singing, and structured therapies enabled children to channel their emotions constructively, highlighting the value of incorporating creative outlets into mental health interventions. Regular physical activities, such as swimming, cycling, hydrotherapy, and structured physiotherapy not only addressed physical development but also boosted children's mental health and confidence:

> "In the swimming pool…the sheer freedom on her face is just wonderful." (#230)

Some parents said that consistent access to mental health support, such as CAMHS or tailored therapies, helped children's emotional regulation, self-esteem, and overall well-being. For those with individually tailored care plans, significant stabilisation of mental health was reported. This was often accompanied by better routines, greater independence, and engagement in enriching activities.

> "His mental health was brilliant…He was happy, up and showered, and ready to go at 8:30." (#201)

When SEND provision was in place, parents reported that their own stress levels also reduced because their child was more settled and sleeping better.

**Gaining autonomy and independence.** SEND provision significantly contributed to children developing foundational life skills such as money management, cooking, shopping, and personal hygiene. These structured routines helped foster autonomy and prepared children for adult responsibilities. For example, one child showed progress in completing basic tasks such as washing clothes, ironing, and planning meals. However, challenges remained in areas such as road safety, due to limited awareness of danger, highlighting the ongoing need for mobility and safety support:

> "She doesn't have it, she can't, she just can't retain that knowledge of even the basic rules of crossing the road." (#221)

The same parent noted that interventions such as residential care settings, tailored educational plans, and activities that promoted autonomy were highlighted as particularly impactful.

> "Since she's moved out, she is much more her old self. Much more settled. Much happier, much more content." (#221)

Transitions to supported living were highlighted as a crucial step for independence. Children showed increased reliance on structured routines and support systems, which reduced dependency on parents and fostered self-reliance:

> "He's getting more and more independent being where he is at the moment because he's not relying on us to do the things for him." (#224)

Several parents highlighted the value of successful internships and work placements in providing young people with a sense of meaning and purpose. Positive experiences were reported where SEND provision included supported internships, often facilitated by third sector organisations and businesses. These opportunities enabled children to demonstrate growth and potential through tailored work experiences.

> "It's not so much about her actually being able to do something. It's what it gave her, she was really proud, that she was contributing, and that she was a part of it." (#221)

**Theme 4: Negative outcomes following SEND provision**

   **Child mental health crises from delayed, inconsistent or prematurely removed provision.** Many parents spoke about how the school environment had exacerbated their child's anxiety, leading to meltdowns, physical illness, and behavioural issues. Negative consequences of inappropriate, poorly implemented, delayed or removed provision included anxiety, suicidal ideation, and eating disorders. These challenges persisted even in specialised settings, such as autism-focused units. Practices such as internal exclusion were sometimes used to manage behaviour and segregate students; however, parents felt that these measures failed to address the root causes. In one case, it contributed to a deterioration in the child's mental health, leading to self-harm and school refusal.

   *"It was just trauma inflicted from start to finish… How can they say everything's okay when the child is suicidal or self-harming?" (#210)*

   Several children had experienced prolonged isolation and trauma prior to receiving tailored support. For some, anxiety significantly impacted their physical health, manifesting as stomach issues, chronic fatigue, or other stress-related symptoms.

   *"Physical health was deeply affected by her anxiety, with symptoms like sleeplessness and germ anxieties." (#236)*

   **Lost education from slow or inappropriate decision-making.** Many parents had successfully appealed against LA decisions, including refusals to assess children for an EHCP, or the naming of schools that could not meet a child's needs. However, the appeals process involved lengthy delays, during which children missed out on valuable education. In some cases, this resulted in months or even years without any learning support and being non-electively home educated. These difficulties in accessing appropriate SEND provision led to disrupted education, lost learning opportunities and lasting trauma.

   *"…that's when problems began because, they were, they were delaying making decisions about what support [CYP NAME] could have [what they decided was] not at all suitable for her needs…So she missed the whole term simply because they were so delayed in processing it." (#235)*

   Exclusionary practices, such as off rolling and chronic unmet needs, were recurring themes. Even when provision was agreed upon, its implementation was often delayed leading lost time and opportunities.

   *"The support that she has within her EHCP was not in place….she could have had a much better time from, for that year where, you know, it took me to go back to the process, that those 18 months, we could have potentially made strides of progress if she had had the EHCP and that additional support." (#247)*

   **Education policies undermine children's health and academic potential.** Many parents believed that the national curriculum, pedagogical methods, ways of assessing achievement and attendance expectations were too rigid and narrow, often harming children's potential rather than highlighting their abilities.

   *"It [dyslexia] impacts you, not just every day, not just every minute, but every second…focus in our mainstream schooling, is on reading and writing. It's the way that we do it. And she's assessed in ways that we already know she's going to struggle." (#241)*

   Another parent said that attendance policies can punish children and parents, if the setting was not appropriate. Parents were "scared" to break the law and keep their children at home, facing a difficult "trade-off between supporting your child and understanding them".

*"[one child said to their parent they had considered suicide because they] know it's law that children have to go to school, and I don't want to get you into trouble." (#201)*

National policy promoting inclusion of children with SEND in mainstream settings was perceived to be influencing professionals' decisions about educational placements.

*"The educational psychologist system seemed to be very determined to advocate for children with special needs to stay in mainstream school." (#235)*

**Professionals lack aspiration and long-term strategies, limiting children's potential and independence.** Many parents noted that limited understanding of professionals directly impacted the opportunities and aspirations for children with SEND, as well as their treatment and provision. One parent recounted an experience where a professional, at the time of her child's diagnosis, suggested that she limit her aspirations for her child. Despite this, the child later developed a deep knowledge and interest in engineering, thanks to the encouragement of their parent. The professional reportedly said:

*"…stick him in a diagnostic nursery make the use of social care, get some respite and get on with your life, he'll never amount to anything." (#201)*

Many parents were worried about the lack of aspiration for children with SEND, noting that this was misguided and negatively impacted long-term outcomes.

*"If you have no aspirations for the students, you're not gonna find out what they're capable of and I think that's, that's their mistake they assume they can't do it because they are, they're autistic or they have got Down syndrome, and they struggle a little bit with communicating." (#224)*

Another parent was informed by the school that they would not support her appeal against her child's borderline result in the school entrance exam:

*"They said yes he's a clever boy, but he's not going to make it…and I said…you have to push him…you have to be Sergeant Major the whole time, which is draining but…that works for him and he [the teacher] said…if he goes to an 11 plus grammar school, that needs to be in him. The teachers are not going to say come, come, come you need to do it." (#233)*

Most parents expressed significant concerns about their children's futures. Many acknowledged that ongoing support would be necessary but expressed doubts about the adequacy of future health and care systems as their children transition to adulthood. The absence of robust post-education support systems posed a risk of social isolation, regression, and missed opportunities for gaining independence:

*"… this generation of children growing up…home and forgotten. And you know, nobody really helps you, once your child has left education, and that is a real, a real hurdle. And it's very worrying and I think it causes huge health problems and depression, for the child, or adult then, and as well as the parents, and parents I know that for a fact, and that is a serious problem that the government need to do something about." (#230)*

While some children made significant progress in independence through consistent SEND provision, others faced stagnation due to the lack of tailored, ongoing support and poor transitions, which led to a reversal of their progress. Assistive tools and embedded coping mechanisms from an early age were identified as essential in reducing risks of burnout and fostering long-term autonomy:

*"When they are in moments of anxiety or difficulties, they have a toolbox of things to support their life, rather than being in burnout." (#234)*

Most parents said they wanted their children to achieve as much independence as possible. This goal was linked to reducing reliance on adult services, improving future mental wellbeing and contributing meaningfully to society. Several parents noted that LAs were not adequately preparing young people with SEND for adulthood, leaving parents to fill the gaps:

*"If you go to higher education, you no longer need an EHCP, effectively [LA] Council see this and say right tick him off, we don't have to worry anymore. He fits into this new category of DSA support. And the continuum there was very weak…we did submit all the EHCP stuff to them. But that was because we took the initiative to do it. So, if we hadn't, then it wouldn't have been there you know, so that would have just stopped." (#209)*

A recurring concern among many parents was the fear of who would advocate for and oversee their child's needs after they were no longer able to do so. They expressed deep worry for their child's future, when they would no longer be able to monitor inputs, and advocate on their child's behalf and provide additional supports not provided by the LA. A parent described how their child could "fall in a pit" if they were not there to fight for their support:

*"We'll not be able to look after her forever." (#211)*

Many parents expressed a desire for increased supports in the workplace to allow adults with SEND and exceptional abilities to contribute. Suggestions included giving clearly defined jobs, flexible working environments tailored to individual needs, and reduced emphasis on interpersonal skills. Some parents believed that businesses as future employers, share responsibility for fostering independence and inclusion of adults with SEND. One parent, actively engaged with businesses to advocate for and promote understanding of her child's employment:

*"We're changing the way businesses think for hidden disability. And people like [CHILD'S NAME]. So, you know, it's good for [CHILD'S NAME], but it's also paving the way for others and getting businesses to change their attitude and support uniquely." (#201)*

**Parents' mental health, capacity to work and family finances are harmed by the system.** Parents frequently described the process of seeking SEND support for their child as traumatic for both them and their child, often leading to a distrust of professionals, particularly when their child's needs were not met. Several reported that they were not listened to until there was a crisis. Parents felt "shut out" and not involved in meetings or decision making and noted that minutes of meetings did not accurately reflect their contributions. One parent described how she was made to feel guilty for requesting respite and the costs involved (#201). Some parents described feeling directly threatened when challenging the system, citing examples of section 47 assessments or safeguarding investigations being triggered by communication breakdowns with children's social care and LAs (e.g., #221). This had a significant impact on parental health:

*"My health hasn't been great, the, the stress on me. I ended up having a breakdown... I ended up in in hospital for two months." (#223)*

Many parents described substantial personal and financial losses associated with securing SEND provision. Some were forced to leave employment due to the high level of need that their child required, and the administrative burden of navigating assessments, appeals and provision processes. One parent reflected on financial toll of leaving her job, losing

pension contributions and depleting tax credits – describing the impact as financially devastating. Time spent managing ongoing communications with schools, LAs and health and care systems was noted, with appeals and tribunals described as particularly time consuming; they "wasted a lot of time" (#226). Several reflected that LAs used significant resources to defend appeals and tribunals which could be better directed toward supporting children with SEND.

Several parents reported having to privately fund independent assessments which were not provided by health or education to identify their child's additional SEND. Most cited assessments were for Speech and Language needs and Dyslexia. A parent said funding reports and tutors over the years had cost them "thousands and thousands of pounds" (#241).

## Discussion

### Summary of key findings

Parents play a major role in advocating for their child and facilitating support. However, it takes a toll to fight with the system, emotionally and financially. This also means that children without this advocacy or with parents who are unable to fight or navigate the system may be losing out and not getting the support they need, increasing societal inequalities. These findings are echoed in other literature from the UK [19,20]. Legal protections are important to protect the rights of children, but the legal system is slow and may be used by LAs to delay provision and reduce expenditure. There is a stark contrast between parents who sacrifice time, money and sometimes employment prior to a tribunal, and LAs who don't prepare anything in advance and concede on the day – this is suggestive of a tactic and deepens parents' mistrust of LAs; it can only be addressed with stronger accountability mechanisms within the system.

Effective communication from professionals is highly valued by parents but is often inadequate, lacking clarity and further builds mistrust. These findings have been identified in several UK studies [31,40]. There are also gaps and persistent delays accessing CAMHS, which are having detrimental long-term effects on children and families. CAMHS provision, including waiting lists and children's outcomes are known to be unacceptable [41]. Strengthening mental health provision and increasing resources should be a priority.

Many children with SEND will eventually transition to adult services (health and social care), which appear to be under-resourced and preparation for adulthood happens too late. Nearly all parents expressed extreme worry about the lack of a long-term strategy, and for their children in the future, when they are no longer there to advocate for them. Focusing on building independence, autonomy and belonging, as well as engaging businesses and continuing to expand internships/supported jobs will harness the true potential of young people with SEND. It may also prevent these children – in the words of three parents – becoming adults who are "at home, forgotten", suicidal or self-harming, or the victims of others who exploit their SEND-related vulnerabilities.

### Implications for policy and practice

Our findings aim to inform and improve provision for children with SEND and their families. Professionals working with SEND learners and their families can either be a barrier or facilitator to support. This highlights the clear need for professionals to be trained and have a good understanding of SEND, the support available and how to facilitate families access to that support. This is particularly relevant for children with autism and communication difficulties. Training could positively change attitudes and improve communication between professionals and families, and treatment of children. Professional health and education bodies need to think about incorporating more training, which matches recommendations from other studies [42].

Long-term strategies are needed for children and young people with SEND as they transition to adulthood, to protect their physical and mental health, promote their belonging and harness their potential as active members of society. There is both an economic and moral case for investment. We still await results of investments by the previous government in provision for young people with SEND who are aged 16–25 [43]. Despite this age-group being eligible for SEND provision, their access has been curtailed by lack of funding, poor transitions, and lack of strategic and regional planning by LAs [44,45].

Future SEND reforms should carefully consider how to strengthen accountability mechanisms to encourage child-centred practice and reduce the need for parents to take LAs to costly tribunals which cause delays for children's provision. Choice of education settings are a clear cause of conflict between parents and authorities, with the latter basing decisions more on cost, the former on the ability of a setting to meet their child's needs. Funding versus needs-based conflict will continue without clearer guidelines on this, and more resources for mainstream and specialist settings. Current short-termism within LAs about annual expenditure blocks the broader view, longer-term outlook and potential cost-savings of fair and just provision for SEND learners now.

**Ideas for future research**

1. How can parental advocacy and collaboration with educators be better channelled to support children's educational and life outcomes?

2. How might earlier, tailored education and mental health interventions prevent educational disruption and improve outcomes for children with SEND?

3. How can providers ensure continuity of SEND-related support during transitions within education and into adulthood, and incorporate long-term strategies to maintain life skills and reduce regression post-education?

4. To what extent should creative and community-based activities be integrated into SEND frameworks, and expanded to address social, emotional, and mental health needs and independence?

5. What are the long-term education and health trajectories and outcomes for young people with SEND?

6. What do long-term health economic analyses reveal about the costs of SEND interventions offset against later cost savings in terms of young people becoming NEET, involved in the criminal justice system, and developing poorer health and mental health?

**Limitations**

Our participants self-selected into the study, and their experiences may not represent the whole diverse SEND population. Whilst we did speak to parents whose children had a range of needs overall, several parents had autistic children and were home educated so we cannot extrapolate our findings.

**Case examples using narrative portraiture**

**Narrative portrait 1: Foster parent of a child with foetal alcohol syndrome**

Primary school: Their foster daughter moved in shortly after starting primary school and when social services "stepped in." She arrived with an EHCP, and specialist school placement arranged after a short spell in mainstream, which was "a nightmare …she would take her clothes off and just run around in the classroom". Her behaviour and diet were "out of control" due to ADHD, global developmental delay and a poorly managed auto-immune condition. Subsequent review meetings with health, education and social services were anchored around the EHCP: "It was a target set…you had an action plan…the social workers would bring along to the looked after review or to the care plan meeting…it's written down as well that she's got to have it done". The specialist primary school was "brilliant" and "very very supportive" with "small classes, lots of staff… top notch". She "had people around her, who…knew her inside out…the building block for her going into secondary." The school offered a parenting course about ADHD which "made such a difference" when it was applied at home.

Secondary school: She had a "really good transition" to a secondary special school for learning disabilities. The structure and timetables were helpful for her. The school set many "aspirations" for her future. Arranging transport was "a right

carry on" between city and county services about who would fund it. The problem was resolved two weeks before school began "…kind of worrying…when you're a parent…you're thinking, what will happen on the Monday, do we take her in?"

The participant praised school staff for clear "strategies" to deal with challenging behaviours recognising the "trigger point they divert them to something else" There has been continuity attending the same school for five years and "regular meetings" "they're always in touch.". Her ADHD medication "calms" her ("her behavior can go from a 100 miles an hour to [laughs] 300 miles an hour"). Her diet and weight are well controlled with a joined-up approach between school and home. Many health, social and SEND services and reviews she needs take place at school, or a short walk away for mental health support. Their experience has been "good". They rarely wait for appointments. Professionals attend meetings as a "priority" "because she's a looked after child, they have a statute of responsibility…to make sure that she's well, looked after, safe. And also that we are as well."

Now and in the future: Sixth form is more "laissez-faire" and "she struggles a bit with" but enjoys the "freedom". She is in a small class at an "excellent school, excellent support" which the participant "cannot fault". Her involvement in team sports have positively impacted her health. She is engaged with CYP mental health services "So, she can talk about what's happening". Her mental health has a "spiky profile". They have an advocate who spends the day with her at weekends, from the city council, who is "very sociable", "plausible". The participant hopes she "can have a good voice for her… independent." The participant thinks she "will continue to need mental health support throughout her life: "I wouldn't be surprised because of her family history there's a little bit of mental illness" They worry there would be damaging waiting times for adult counselling. They "need to talk to somebody now, not six months time." The participant remains a strong advocate: "I attend every meeting …I don't mind arguments with social workers if I don't think it's right."

The participant thinks "the transition to adulthood could be earlier and clearer "it's very, very softly, softly…you're a child and then suddenly, you've got an adult social worker the next day" who [you've] "never met". At every meeting "I say, what's happening with adulthood?...Where's the worker?" The participant worries that "I'm getting older, my wife's getting older we'll not be able to look after her forever. There should be a good planned transition." They see it as a foreshadowing of future difficulties accessing adult care with "fragmented" social care teams, "stretched at breaking point", with "new people" and "agency staff" who are still "building the picture". "We are lucky because she's looked after, if she wasn't looked after, would it be quite as important?…what do we do with complex children, who are going into adulthood, there's no bridging at all…if we weren't there batting for them. Who else would?". The participant thinks if she didn't have that support she would "go downhill rapidly…she needs to have that support mechanisms in place…if support is not there, she'll follow the wrong crowd…with her capacities the issue, she wouldn't know she's being taken advantage of."

### Narrative portrait 2: Parent of young teenager, with autism and other needs, home-educated

Primary School: His son "didn't want to go to school" and was not able to "identify why". He found it difficult to express himself and the only thing he could say was that he "hate[d] wearing the school uniform" and that his "jumper didn't feel right". When his son was aged 5 years he came home with "a little slip" saying that he "was on the SEN Register". After this there were several professionals involved in assessing his son's needs, but discussions seemed to "go round in circles" and recommendations were not implemented. The school were reluctant to "make reasonable adjustments" like allowing him to "keep his PE kit on" when he found that soothing or making "his ear defenders" readily available. Some of the professionals did not believe the descriptions he gave about his son's difficulties and he and his wife came "under investigation for fabricated and induced illness". He "would go into a room with[…seven or eight professionals…all grouped together and [him] sat in a space by [himself]…it was awful". Once his son "was diagnosed with sensory processing disorder…they stopped the investigation" but "it was probably the worst year of [his family's] life". His son "started to enjoy school" in Year 5 when he had a class teacher who had personal experience of autism and "she was really lovely with them" but "nothing was formalised". He thought "the thing that proved [they] could home educate…was…having a national lockdown". His son improved during COVID because they "spent more time together" and it "took the pressure

away". He began "to eat most things", and his diet expanded so that now "he doesn't see a dietitian anymore". However, when he "returned to school [they] saw [things] revert" so they "pulled him out" in year 6. Home schooling is "probably… one of the best things [they have] ever done for him [and] friends and family noticed a big difference with him as soon as it happened as well".

Secondary School: His son has been "home educated" for the past 3 years and "really enjoys [it]". He and his wife home educate together "Monday to Friday, between 9am and 3pm, [because] that kind of structure works for him". They take a "child led" approach so that his son "gets to learn about the topics, that he wants to learn, as well as…English and maths". They focus on "life skills" and his son is "engaging really well [because] it's kind of tailored around him and it's very flexible". His son has a "passion for…animals, especially…marine life and…birds", so they visit a local "bird of prey [sanctuary], and…an aquarium" as part of his learning. His son did not receive his Education Health and Care Plan (EHCP) until last year but as part of this he sees a psychotherapist for "the [school related] trauma" and a "sensory integration therapist". His son "does melt down [and] he get[s] upset and overwhelmed …but [he is able] to use tools to regulate himself" now. The "sensory integration therapy…, psychotherapist…and just the home environment together… have improved his health and well-being". They asked the local authority (LA) if they could manage the "EHCP personal budget" to allow them to "do that bit extra with [their son], like… go somewhere [educational]…every week [and pay for] travel…but [the LA] said no".

Now & in the future: Looking towards the future if his son wanted "to sit some GCSE's" he would support him to do that. In five years, his son would be aged 18 years and might go to college "if the college [could] meet his needs". His daughter, who also has SEN, was well supported there so he was hopeful that "the whole [college] culture might be good for" his son too. He thought that "having …independence at that age… might be good" for his son but it was "hard to envisage… now". He thought his son "might need a teaching assistant in the classroom… that would be independent from [his] parents" and that this would "hopefully help him transition towards being more independent". He thought with "the correct support [his son] could do a lot". Finding "the correct support" depended "on the decisions of other people such as [the upcoming] tribunal judge, the SEN panel, [other] professionals' reports". Thinking about the future was frightening but he hoped his son would "get into something that he wants to do…be the best he can…and live up to his potential".

### Narrative portrait 3: Grandmother of a primary-school aged child with autism

Early Years: Looking back, she could see her grandson showed "signs of autism" as he "would never interact at toddler groups" and "made it disruptive for other children", "he... didn't toilet train in the time periods that children normally would do" and "he didn't crawl". At about 13 months "he wasn't really getting any interaction with other children" so she enrolled him at nursery school.

Nursery School: Staff observed that "he was so bright" because he had "some speech…knew all his colours [and] could count to 10" but she noticed "his language didn't improve, [and] he didn't really make any friends". She thought he was "not progressing as he should be" and as there is a family history of autism, she "raised her concerns with the nursery" and "pushed [for] an autism referral". He was aged 2 years at this point. The local authority delayed the referral but "she wrote to them and "started ringing around...saying 'someone's going to have to do something here'". As a result "they… decided that nursery would do the speech and language therapy referral, and…the health visitor would do the autism referral". However, the process became very protracted and "by the time got his autism diagnosis...he was in school". Primary School: They selected a "small rural [primary] school" with "small classes" so it would be less overwhelming for him. She was "upfront" with school "that he was...being assessed for autism" and they were "surprised that his issues hadn't been picked up and addressed sooner". She thought "they felt he should have come into school with a plan". There was a "period of a couple of months [...] trying to get information from the nursery passed across, and that was problematic". She found "his teacher at school was...proactive" so that "in January [of reception] they confirmed they would assess him" and "the plan was...issued in....April". Currently "he's operating...as a three [or] four-year-old [and] he has quite

severe speech and language difficulties, mainly around comprehension" which affects his engagement with the curriculum. He receives "full-time one to one support" from a "classroom assistant" and "he is seen...once a quarter by a speech and language therapist, who... then makes recommendations [for] the next term". They leave it "to the teaching assistant to deliver the recommendations". She thought there would be...learning strategies [specifically] for autistic children" but she was "not sure that [was] happening" because he had "a general teaching assistant". She thought he was "making progress but" expected it "would be quicker than it is at the moment". She thought "that he [was] not really getting the support that he should be getting at the moment because the budget's just not there".

Now & in the future: In five years "he will be...making the transition from primary school to secondary school" which she thought would "be a massive transition for him". She hoped he would be as "prepared as possible for that transition" and "that his speech and language difficulties" would "have been largely addressed". She wanted him to continue "to develop socially and [be] able to form and maintain friendships". She wanted "a clearer path for his independence, as he heads towards being a young adult". She liked "to think that" the support he has received "will have...had a positive impact on his mental health" as "they [were] building a good platform for helping him improve "social interactions" with his peers. Looking beyond 5 years "he has a condition, that's not going to go away" and she wanted to ensure that he is "supported...to live a healthy and fulfilling life". She would "continue to advocate for him until he" no longer needed her to. She used to work for local authorities" and knows her "way around the system [and] which buttons to press to get things done". She thought it was "unfair" that the service parents get for their children is to do with the "parents' ability to advocate for their child".

## Supporting information

**S1 File. Supplementary file.**
(DOCX)

## Acknowledgments

We would like to express our gratitude to all the children and families who took part in this research.

We would also like to thank Ananya Khera for her assistance in transcribing the interviews, our advisory groups for guiding the study, the HOPE Study Steering Committee: Chris Bonell, Kate Evans-Jones, Julia Ogden, Jo Hutchison, Karen Horridge, and the wider HOPE Study consortium for their support and input throughout.

The HOPE Study consortium includes the following members: Ruth Gilbert[1] (PI and consortium lead; r.gilbert@ucl.ac.uk), Katie Harron[1], Bianca L De Stavola[1], Lorraine Dearden[1], Tamsin Ford[2], Kate Lewis[1], Vincent Nguyen[1], Ania Zylbersztejn[1,3], Jennifer Saxton[2], Jacob Matthews[2], William Farr[4], Ayana Cant[1], Laura Gimeno[3], Isaac Winterburn[2], Ariadna Albajara Saenz[2], Andrea Aparicio Castro[5], Julia Shumway[1], Lucy Karwatowska[1], Ananya Khera[2], Nicolas Libuy[6], Louise Macaulay[1], Matthew Lillimam[1], Kate Boddy[7], Stuart Logan[7], Jugnoo Rahi[1,8], Kristine Black-Hawkins[2], Johnny Downs[9].

Affiliations:

1 University College London Great Ormond Street Institute of Child Health, London, WC1N 1EH

2 Department of Psychiatry, University of Cambridge, Cambridge, CB2 0SZ

3 NIHR Great Ormond Street Hospital Biomedical Research Centre, Great Ormond Street Hospital for Children NHS Foundation Trust, London, WC1N 3JH

4 Faculty of Education, University of Cambridge, Cambridge, CB2 8PQ

5 Leverhulme Centre for Demographic Science, University of Oxford

6 Centre for Longitudinal Studies, Social Research Institute, University College London, London, WC1H 0NU

7 Department of Health and Community Sciences, University of Exeter, Exeter, EX1 2HZ

8 University College London Institute of Ophthalmology, London, EC1V 9EL

9 Institute of Psychiatry, Psychology and Neuroscience, Kings College London, SE5 8AB

## Author contributions

**Conceptualization:** Jennifer Saxton, Jacob Matthews, Isaac Winterburn, Sarah Barnes, Kristine Black-Hawkins, Tamsin Ford.

**Data curation:** Jennifer Saxton, Anne-Marie Burn, Xinhe Zhang, Hilary Toulmin, Jennifer Parker, Helen Casey, Jacob Matthews, Isaac Winterburn, Charlotte Tripp, Sarah Barnes, Poppy Hall.

**Formal analysis:** Jennifer Saxton, Anne-Marie Burn, Xinhe Zhang, Hilary Toulmin, Jennifer Parker, Helen Casey.

**Funding acquisition:** Kristine Black-Hawkins, Tamsin Ford.

**Investigation:** Jennifer Saxton.

**Methodology:** Jennifer Saxton, Anne-Marie Burn, Kristine Black-Hawkins, Tamsin Ford.

**Project administration:** Jacob Matthews, Isaac Winterburn, Sarah Barnes.

**Software:** Jennifer Saxton.

**Supervision:** Jennifer Saxton.

**Validation:** Jennifer Saxton.

**Writing – original draft:** Jennifer Saxton, Anne-Marie Burn, Xinhe Zhang, Hilary Toulmin, Jennifer Parker, Helen Casey, Hayley Gains.

**Writing – review & editing:** Jennifer Saxton, Anne-Marie Burn, Xinhe Zhang, Hilary Toulmin, Jennifer Parker, Helen Casey, Kristine Black-Hawkins, Hayley Gains, Tamsin Ford.

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
