## [Decision Letter · Decision Letter 0]

23 Jul 2025

PONE-D-25-23111Barriers, enablers and outcomes reported by parents engaged with the special educational needs system in England: A qualitative studyPLOS ONE

Dear Dr. Saxton,

Thank you for submitting your manuscript to PLOS ONE. After careful consideration, we feel that it has merit but does not fully meet PLOS ONE’s publication criteria as it currently stands. Therefore, we invite you to submit a revised version of the manuscript that addresses the points raised during the review process.

Thank you for your submission. Based on the reviewer’s detailed feedback and my own evaluation, I am recommending a **major revision** at this stage.

The manuscript addresses a relevant and timely topic and meets PLOS ONE’s criteria for methodological soundness and ethical research. However, several key revisions are needed to ensure clarity, alignment of study components, and compliance with journal policies.

**Changes required for acceptance include:**

Clearer alignment between the stated purpose of the study and the research questions, particularly regarding claims about links between provision and outcomes.Reorganization and clarification of the literature review, ideally structured around the stated research questions.Expanded detail about participant demographics and child characteristics.Improved explanation of UK-specific terms (e.g., SEN, EHCP) for an international audience.Relocation of methodological details currently found in the Results section (e.g., interview duration).Clearer presentation of children's needs in the Results section.A revised Data Availability Statement that aligns with PLOS ONE’s policy, including appropriate justification if data cannot be publicly shared.

**Recommended changes (not mandatory for acceptance):**

Enhancing clarity through subheadings within the Introduction.Minor improvements in reporting analytic rigor and theme development.

There are no conflicting recommendations across reviews. The reviewer’s suggestions are constructive and coherent, and I encourage the authors to address them thoroughly in the revision.

Once these issues are addressed, the manuscript will be suitable for reconsideration.

We look forward to receiving your revised manuscript.

Kind regards,

Ramandeep Kaur

Academic Editor

PLOS ONE

Journal Requirements:

[This research was supported by Health Data Research UK under grant No. LOND1), which is funded by the UK Medical Research Council and eight other funders. TF is supported by a National Institute of Health Research (NIHR) senior investigator award (https://www.nihr.ac.uk/). All research at the Department of Psychiatry in the University of Cambridge benefits from the NIHR Cambridge Biomedical Research Centre (NIHR203312) and NIHR Applied Research Collaboration East of England. This study/project is funded by the National Institute for Health Research (NIHR) under its Programme Grants for Applied Research Programme (Grant Reference number NIHR- NIHR202025). The funders did not play any role in the study design, data collection and analysis, decision to publish, or preparation of the manuscript. The views expressed are those of the author(s) and not necessarily those of the NIHR or the Department of Health and Social Care.].

a) If there are ethical or legal restrictions on sharing a de-identified data set, please explain them in detail (e.g., data contain potentially identifying or sensitive patient information, data are owned by a third-party organization, etc.) and who has imposed them (e.g., a Research Ethics Committee or Institutional Review Board, etc.). Please also provide contact information for a data access committee, ethics committee, or other institutional body to which data requests may be sent

5. One of the noted authors is a group or consortium [The HOPE Study]. In addition to naming the author group, please list the individual authors and affiliations within this group in the acknowledgments section of your manuscript. Please also indicate clearly a lead author for this group along with a contact email address.

6. Please upload a copy of Supporting Information Figure/Table/etc. (4.0 COREQ checklist) which you refer to in your text on page 46.

Additional Editor Comments:

Thank you for submitting your manuscript entitled "Barriers, enablers and outcomes reported by parents engaged with the special educational needs system in England: A qualitative study." The paper presents important insights into the lived experiences of parents navigating the SEND system in England.

Based on the comments from Reviewer 1, I am requesting a major revision of your manuscript. While the overall study is well-conceived and contributes to the literature, the reviewer has raised several concerns that must be addressed to strengthen the manuscript:

Clarify the alignment between the study’s purpose, research questions, and findings. The reviewer noted that the stated purpose (examining the link between provision and outcomes) does not align well with the qualitative nature of the study, which explores reported experiences rather than explicit causal links.

Improve the organization of the literature review. Please consider restructuring this section around your two research questions, possibly using subheadings to enhance clarity and coherence.

Clarify terminology and context. Please spell out acronyms (e.g., SEN) at first mention and provide brief explanations of UK-specific terms like EHCPs to aid international readers.

Expand on participant details. Include additional demographic and contextual information about participants (e.g., gender, caregiver role, whether children were home-educated).

Relocate interview duration detail. This currently appears in the Results but should be moved to the Methods section.

Clarify the description of children’s needs. Reformat this information for clearer presentation and understanding.

Address the data availability policy. As your current statement does not meet PLOS ONE’s data sharing requirements, please revise it to explain any ethical or legal limitations on data sharing and provide what is possible within those constraints.

Please revise your manuscript accordingly and include a detailed point-by-point response to the reviewer comments.

Reviewers' comments:

Reviewer's Responses to Questions

**Comments to the Author**

1. Is the manuscript technically sound, and do the data support the conclusions?

Reviewer #1: Yes

2. Has the statistical analysis been performed appropriately and rigorously? 

Reviewer #1: N/A

3. Have the authors made all data underlying the findings in their manuscript fully available?

Reviewer #1: No

4. Is the manuscript presented in an intelligible fashion and written in standard English?

Reviewer #1: Yes

5. Review Comments to the Author

Reviewer #1: Brief Summary

This manuscript presents the results of a qualitative study in which the authors examined “the experiences of 22 parents in identifying their child’s needs, securing provision, and its impact on the child and their family.” Results suggests that “Thematic framework analysis identified legal protections and the advocacy efforts of parents and professionals as key enablers. Barriers included professionals’ lack of understanding about SEND (particularly autism), poor communication between services and families, and system failures. Positive outcomes from SEND provision included improved health, education and social outcomes, and young people gaining autonomy and independence. Negative outcomes included lost educational opportunities, worsening mental health for both children and their parents, and educational policies lacking long term vision.”

General Comments and Overall Evaluation

This study is important because it adds to the literature that examined parents’ experience with their child’s Special Educational Needs and Disabilities (SEND) services. Overall, the study described in this manuscript examines these issues in a way that might be of interest to the readers of PLOS ONE. However, the manuscript would benefit from a better alignment between the purpose of the study, the research questions, and the results, as well as a more organized literature review. I summarize below my comments and suggestions along these lines.

Introduction

The review of research literature was not well organized. The authors may want to consider organizing the literature review according to the two research questions and adding subheadings to make it easier for readers to follow.

The identified gap in literature does not align with the purpose of the study. The authors stated that “few peer-reviewed studies have examined the link between SEND provision and children’s and families health, education and other outcomes, or the antecedent barriers and enablers of identifying SEND and securing appropriate provision.” However, the current qualitative study did not address this gap (the link between these variables).

Research Question 1: Please spell out SEN.

The research questions are clearly stated. However, the manuscript would be strengthened if the purpose of the study aligns with the two research questions.

Methods

Participants: The manuscript would benefit from more descriptions about the participants and their children. For example, are participants parents and/or grandparents? Mothers or fathers? How many children were home-educated?

Data Collection: The authors may consider moving the information on the interview duration (2.5 hours) to the Data Collection section in the Methods from the result (p. 11, line 245).

Data analysis was clearly described and appropriate.

Results

Line 247, please provide more explanations on this statement “Nineteen children had EHCPs or a SEN statement and three were receiving SEN support” to provide a context for readers from other countries.

Line 251-254: “Parents described their children’s SEND as including at least one of the following: autism (n=13), several having further needs such as ADHD and hypermobility, a mental health problem (n=5), a learning disability (n=5), a brain injury (n=4), and speech and language difficulties, some of whom were non-verbal (n=5).” This information needs to be more clearly presented.

Four main themes were identified and within each main theme, subthemes were clearly presented in Table 1. Results were systematically presented.

I hope that these comments are helpful.

6. PLOS authors have the option to publish the peer review history of their article (what does this mean? ). If published, this will include your full peer review and any attached files.

**Do you want your identity to be public for this peer review?** For information about this choice, including consent withdrawal, please see our Privacy Policy .

Reviewer #1: No

---

## [Author Response · Author response to Decision Letter 1]

19 Sep 2025

No. Commenter Our Response

Editor

1 Please ensure that your manuscript meets PLOS ONE's style requirements, including those for file naming. • To the title page we have added the Department of Psychiatry address, and a footnote explaining that the full list of HOPE consortium members is available in the acknowledgements.

• In the main manuscript we have modified the font sizes and emboldened the three heading levels permitted.

• On page 50 of the main manuscript we have replicated the naming conventions for supplementary information captions and supplementary files, based on the PLOS One guidelines.

• We have renamed the supplementary file as ‘S1 file. Supplementary File’.

2 We note that the grant information you provided in the ‘Funding Information’ and ‘Financial Disclosure’ sections do not match. When you resubmit, please ensure that you provide the correct grant numbers for the awards you received for your study in the ‘Funding Information’ section.

• Having searched the file inventory and original submission pdf we are only able to see the text we provided entitled ‘Funding Statement’.

• We can confirm (and clarify) that the NIHR award number for The HOPE Study was listed correctly (NIHR202025), and there is a separate general funding award which supports all our department’s work (NIHR203312). Perhaps this was identified as a mismatch of grant numbers, but they are both relevant and supported this study.

• I have copied the funding statement (amended in line with comment 3 below and reordered for greater clarity) into the cover letter as requested.

3 Thank you for stating in your Funding Statement:

[This research was supported by Health Data Research UK under grant No. LOND1), which is funded by the UK Medical Research Council and eight other funders. TF is supported by a National Institute of Health Research (NIHR) senior investigator award (https://www.nihr.ac.uk/). All research at the Department of Psychiatry in the University of Cambridge benefits from the NIHR Cambridge Biomedical Research Centre (NIHR203312) and NIHR Applied Research Collaboration East of England. This study/project is funded by the National Institute for Health Research (NIHR) under its Programme Grants for Applied Research Programme (Grant Reference number NIHR- NIHR202025). The funders did not play any role in the study design, data collection and analysis, decision to publish, or preparation of the manuscript. The views expressed are those of the author(s) and not necessarily those of the NIHR or the Department of Health and Social Care.].

• Please provide an amended statement that declares *all* the funding or sources of support (whether external or internal to your organization) received during this study, as detailed online in our guide for authors at http://journals.plos.org/plosone/s/submit-now.

• Please also include the statement “There was no additional external funding received for this study.” in your updated Funding Statement.

• We believe all the funding sources for The HOPE Study have now been declared. For completeness we have included the ECHILD funding (which enabled other HOPE Study quantitative analyses). We originally omitted these from the funding statement because ECHILD data played no role in this qualitative sub-study.

• We have included in the funding statement: “There was no additional external funding received for this study.”

• We have copied the amended funding statements into the cover letter.

4 We note that you have indicated that there are restrictions to data sharing for this study. For studies involving human research participant data or other sensitive data, we encourage authors to share de-identified or anonymized data. However, when data cannot be publicly shared for ethical reasons, we allow authors to make their data sets available upon request. For information on unacceptable data access restrictions, please see http://journals.plos.org/plosone/s/data-availability#loc-unacceptable-data-access-restrictions.

a) If there are ethical or legal restrictions on sharing a de-identified data set, please explain them in detail (e.g., data contain potentially identifying or sensitive patient information, data are owned by a third-party organization, etc.) and who has imposed them (e.g., a Research Ethics Committee or Institutional Review Board, etc.). Please also provide contact information for a data access committee, ethics committee, or other institutional body to which data requests may be sent

a) We are not permitted for ethical reasons to share full interview transcripts to external researchers. Firstly, participants did not consent to wider sharing of their transcripts beyond the analysis team. Secondly, the data collected in our study are qualitative, and because they are based around participants’ life timelines they are replete with information that could directly or indirectly disclose participants’ identities. We can invite interested researchers to contact us for reasonable requests for redacted transcripts though.

b) This point appears to refer to clinical data repositories and is not relevant for qualitative data.

• We will amend our Data Availability statement in the submission form as follows:

‘The datasets generated and/or analysed during the current study are not publicly available due to risk of identifying participants, but redacted versions are available from the corresponding author on reasonable request. (jcs230@cam.ac.uk)’

5 One of the noted authors is a group or consortium [The HOPE Study]. In addition to naming the author group, please list the individual authors and affiliations within this group in the acknowledgments section of your manuscript. Please also indicate clearly a lead author for this group along with a contact email address.

• We have actioned this request (see p.46-47), including identifying Ruth Gilbert as the consortium lead and providing her contact email.

6 Please upload a copy of Supporting Information Figure/Table/etc. (4.0 COREQ checklist) which you refer to in your text on page 46.

• We have embedded the COREQ checklist within the revised Supplementary File. In the prior version of the manuscript the COREQ checklist was incorrectly labelled as 4.0, which has now been corrected to 5.0 (p.50)

7 Clarify the alignment between the study’s purpose, research questions, and findings. The reviewer noted that the stated purpose (examining the link between provision and outcomes) does not align well with the qualitative nature of the study, which explores reported experiences rather than explicit causal links.

• The overall purpose of the HOPE Study is to establish the causal links between SEND provision and health/education/other outcomes in children and families. We appreciate that this is a qualitative sub-study, and should not include quantitative language (e.g. about causal attributions) without mentioning that these are perceived, subjective, reported experiences.

• When conducting the study, we ensured that our questioning within the topic guides, and our subsequent coding of the transcripts involved the parent attributing an aspect of SEND provision to an outcome, rather than a general experience or outcome related to their child’s SEND. This qualitative reporting is still an important, valid, and insightful way to evaluate how SEND provision influences outcomes, but requires nuanced presentation to avoid misunderstandings.

• To respond to the editor and reviewer comments to clarify the language around attributing outcomes to provision we have made the following changes to the manuscript:

o Abstract – we have amended the language ’ to clarify the qualitative and subjective nature of the outcomes parents attributed to SEND provision (lines 29 & 36-38).

o Introduction section – we have included our study aim (lines 201-203) which emphasises that our study focuses on parent reported experiences of how provision influenced experiences and outcomes (to avoid suggestion that we have looked at the quantitative/objective linkage of provision to outcomes).

• One of our papers previously cited as under review has now been published, so we have updated the reference. (Matthews et al ref 28)

8 Improve the organization of the literature review. Please consider restructuring this section around your two research questions, possibly using subheadings to enhance clarity and coherence.

• We have included new sub-headings within the literature review section of the introduction (see pages 3-8).

• The subheadings are ordered from barriers (and enablers) through to outcomes, aligning with the order of the research questions.

• We have reordered some of the literature review accordingly

9 Clarify terminology and context. Please spell out acronyms (e.g., SEN) at first mention and provide brief explanations of UK-specific terms like EHCPs to aid international readers.

• We have spelt out SEN in lines 206-207.

• We have also responded to Reviewer 1’s request to expand on what EHCPs and SEN Support mean (see our response to 20) below.

10 Expand on participant details. Include additional demographic and contextual information about participants (e.g., gender, caregiver role, whether children were home-educated).

• We have added further participant detail, including sex of the caregiver, caregiver role, and the numbers of children attending different education settings (including home education).

• At times we have not provided exact numbers, as the minority sub-groups were so small additional precision in our reporting could have compromised participants’ anonymity through indirect identification (lines 308-317).

11 Relocate interview duration detail. This currently appears in the Results but should be moved to the Methods section.

• We have moved this accordingly (see lines 261-262)

12 Clarify the description of children’s needs. Reformat this information for clearer presentation and understanding.

• See our response to 21) below and in the main manuscript (lines 320-327)

13 Address the data availability policy. As your current statement does not meet PLOS ONE’s data sharing requirements, please revise it to explain any ethical or legal limitations on data sharing and provide what is possible within those constraints.

• Please see our response to 4) above.

Reviewer 1

14 The review of research literature was not well organized. The authors may want to consider organizing the literature review according to the two research questions and adding subheadings to make it easier for readers to follow.

• We have included new sub-headings within the literature review section of the introduction (see our response to 8) above).

15 The identified gap in literature does not align with the purpose of the study. The authors stated that “few peer-reviewed studies have examined the link between SEND provision and children’s and families health, education and other outcomes, or the antecedent barriers and enablers of identifying SEND and securing appropriate provision.” However, the current qualitative study did not address this gap (the link between these variables).

• Our study did intend to examine the link between provision and outcomes, in a qualitative manner. Our topic guide and analyses were structured around identifying enablers, barriers and the link between SEND provision and outcomes (please see 7) above for further information), methods section (lines 310-303), results sections (themes 3 and 4), and the topic guide (supplementary file 1).

• To avoid any misunderstanding that we have looked at this in a quantitative manner, we have tweaked the language to clarify that reported links between provision and outcomes are parent reported.

• Please see our response to point 7) above for additional details

16 Research Question 1: Please spell out SEN.

• We have spelled this out in RQ1 (lines 206-207).

17 The research questions are clearly stated. However, the manuscript would be strengthened if the purpose of the study aligns with the two research questions.

• We have added a unifying study aim directly before the research questions (lines 201-203)

18 Participants: The manuscript would benefit from more descriptions about the participants and their children. For example, are participants parents and/or grandparents? Mothers or fathers? How many children were home-educated?

• We have added a sentence to explain that most of the participants were biological parents, but also included foster and adoptive parents (lines 308-309).

19 Data Collection: The authors may consider moving the information on the interview duration (2.5 hours) to the Data Collection section in the Methods from the result (p. 11, line 245).

• We have moved this sentence accordingly (see lines 261-262).

20 Results: Line 247, please provide more explanations on this statement “Nineteen children had EHCPs or a SEN statement and three were receiving SEN support” to provide a context for readers from other countries.

• We have added further explanatory detail to lines 313-318

• ‘Nineteen children had EHCPs or a Special Educational Needs (SEN) statement (both are legal documents outlining plans for children requiring more support than schools ordinarily provide). Three children were receiving ‘SEN support’ (lower level support, largely led and provided for by schools).’

21 Line 251-254: “Parents described their children’s SEND as including at least one of the following: autism (n=13), several having further needs such as ADHD and hypermobility, a mental health problem (n=5), a learning disability (n=5), a brain injury (n=4), and speech and language difficulties, some of whom were non-verbal (n=5).” This information needs to be more clearly presented.

• We have updated the information here, including providing a median, min and max number of SEND conditions (most CYP had more than one) (lines 320-327).

• We have rephrased this paragraph, which is hopefully clearer and includes the denominator of 22 for each reported condition to reduce confusion.

• We have used the enrolment spreadsheet which includes all SEND reported by parents (primary and secondary), so the numbers have changed.

• The previously reported autism figure was incorrect and has been adjusted in the revised manuscript.

---

## [Editor Report · Decision Letter 1]

14 Oct 2025

Barriers, enablers and outcomes reported by parents engaged with the special educational needs system in England: A qualitative study

PONE-D-25-23111R1

Dear Dr. Saxton,

We’re pleased to inform you that your manuscript has been judged scientifically suitable for publication and will be formally accepted for publication once it meets all outstanding technical requirements.

Kind regards,

Ramandeep Kaur

Academic Editor

PLOS ONE
---

## [Editor Report · Acceptance letter]

PONE-D-25-23111R1

PLOS ONE

Dear Dr. Saxton,

I'm pleased to inform you that your manuscript has been deemed suitable for publication in PLOS ONE. Congratulations! Your manuscript is now being handed over to our production team.

Kind regards,

on behalf of

Dr. Ramandeep Kaur

Academic Editor

PLOS ONE